# Working Algorithms and Detection Methods of Autoantibodies in Autoimmune Liver Disease: A Nationwide Study

**DOI:** 10.3390/diagnostics12030697

**Published:** 2022-03-12

**Authors:** Guillermo Muñoz-Sánchez, Albert Pérez-Isidro, Iñaki Ortiz de Landazuri, Antonio López-Gómez, Luz Yadira Bravo-Gallego, Milagros Garcia-Ormaechea, Maria Rosa Julià, Odette Viñas, Estíbaliz Ruiz-Ortiz

**Affiliations:** 1Department of Immunology, Centre de Diagnòstic Biomèdic, Hospital Clínic de Barcelona, Villarroel 170-Escala 4, Planta 0, 08036 Barcelona, Spain; gumunoz@clinic.cat (G.M.-S.); alperezi@clinic.cat (A.P.-I.); ortizdelan@clinic.cat (I.O.d.L.); luzyadira.bravo@salud.madrid.org (L.Y.B.-G.); ovinyas@clinic.cat (O.V.); 2Institut d’Investigacions Biomèdiques August Pi i Sunyer (IDIBAPS), Universitat de Barcelona, 08036 Barcelona, Spain; 3Department of Immunology, Hospital Universitari Son Espases, 07120 Palma de Mallorca, Spain; antonio.lopezgomez@ssib.es (A.L.-G.); rosa.julia@ssib.es (M.R.J.); 4Institut d’Investigació Sanitària Illes Balears, 07120 Palma de Mallorca, Spain; 5Lime Tree Surgery NHS, Worthing BN14 0DL, UK; milagarci@gmail.com

**Keywords:** autoimmune hepatitis, autoimmune liver diseases, autoantibodies, indirect immunofluorescence assays, antigen-specific techniques, primary biliary cholangitis, primary sclerosing cholangitis

## Abstract

Autoantibody detection is the cornerstone of autoimmune liver diseases (AILD) diagnosis. Standardisation of working algorithms among autoimmunity laboratories, as well as being aware of the sensitivity and specificity of various commercial techniques in daily practice, are still necessary. The aim of this nationwide study is to report the results of the 2020 Autoimmunity Workshop organised by the Autoimmunity Group of the Spanish Society of Immunology and to provide useful information to clinicians and laboratory specialists to improve the management of autoantibody detection in AILD diagnoses. Serum samples from 17 patients with liver diseases were provided by the organisers of the 2020 Autoimmunity Workshop and were subsequently analysed by the 40 participating laboratories. Each laboratory used different techniques for the detection of autoantibodies in each patients’ serum sample, according to their working algorithm. Thus, almost 680 total complete patient reports were obtained, and the number of results from different autoantibody detection techniques was >3000. Up to eight different working algorithms were employed, including indirect immunofluorescence assays (IFA) and antigen-specific techniques (AgST). The IFA of HEp-2 cells was more sensitive than IFA of rat triple tissue for the study of anti-nuclear autoantibodies (ANA) associated with AILD. The IFA of a human neutrophil study for the analysis of anti-neutrophil cytoplasmic autoantibodies was not carried out systemically in all patients, or by all laboratories. AgSTs were the most sensitive methods for the detection of anti-smooth muscle/F-actin, soluble liver antigen, liver cytosol-1, M2-mitochondrial autoantibodies, and ANA associated with primary biliary cholangitis. The main differences in AMA detection were due to patients with autoantibodies against the non-dominant epitope of pyruvate dehydrogenase complex. Given that they are complementary, IFA and AgST should be performed in parallel. If there is high suspicion of AILD, AgST should always be performed.

## 1. Introduction

Autoimmune liver diseases (AILD) comprise three differentiated entities, autoimmune hepatitis (AIH), primary biliary cholangitis (PBC), and primary sclerosing cholangitis (PSC), which are characterised by enhanced inflammation and progressive liver fibrosis. Untreated AILD can be life-threatening, and patients may require a liver transplant to survive. Therefore, early diagnosis is essential for the initiation of adequate therapy [1]. In addition to the analysis of liver biopsies and serum biomarkers of liver damage, the detection of AILD-related autoantibodies plays a central role as a diagnostic tool [2,3,4,5,6,7] and is very useful to discriminate between the three major autoimmune liver diseases (AIH, PBC, and PSC) [8].

Autoantibodies associated with AIH include anti-nuclear autoantibodies (ANA), anti-smooth muscle (SMA)/F-actin and anti-soluble liver antigen/liver pancreas (SLA) autoantibodies for type 1 AIH (AIH-1) and anti-liver kidney microsomal type-1 (LKM-1) and anti-liver cytosol-1 (LC-1) autoantibodies for type 2 AIH (AIH-2). PBC is characterised by the detection of anti-mitochondrial autoantibodies (AMA) type-2 (AMA-M2). In addition to AMA, some ANAs are highly specific for PBC, being especially relevant in AMA-M2-negative patients and also for prognosis [9]. The main targets for those ANAs are anti-Sp100 and anti-gp210 autoantibodies. Recently, PBC has also been associated with the presence of anti-hexokinase-1 and anti-kelch-like-protein-12 autoantibodies [10,11]. The sensitivity and specificity characteristics of the different autoantibodies described above, in terms of the clinical relevance of their presence or absence in the diagnosis of the disease, as well as their different immunofluorescence patterns, are shown in Table 1 and Figure 1.

PSC is a chronic, idiopathic, cholestatic liver disease characterised by peribiliary inflammation and fibrosis [12]. Even though immune serology is considered unspecific for PSC [13], several autoantibodies have been described in patients with PSC. In fact, anti-neutrophil cytoplasmic autoantibodies (ANCAs) are more prevalent biomarkers than ANA or SMA for PSC.

Moreover, there are other autoantibodies that are associated (but not specific to) with AILD. ANCAs belong to this group and are included in the updated score system for AIH diagnosis [6]. Atypical ANCA (aANCA), also known as peripheral anti-nuclear neutrophil autoantibodies (pANNA), is the main AIH-associated reactivity pattern. The proposed target of pANNA is tubulin-β-chain-5 [14], a 50-kDa neutrophil-specific protein, but there are no commercial kits available to test for these autoantibodies. Additionally, anti-Ro52 and anti-centromere autoantibodies, although non-AILD specific, have been suggested to be useful in the diagnosis and prognosis of PBC [15] and some AIH guidelines include anti-Ro52 autoantibodies [16]. However, further studies in the search of new biomarkers are still required, i.e., the IgA isotype, which, in addition to helping in diagnosis, have a prognostic value in PSC patients [17,18].

The screening for AILD-associated autoantibodies has traditionally been performed using indirect immunofluorescence assays on rat triple tissue (IFA-RTT) or HEp-2 cells (IFA-HEp-2) and complemented by antigen-specific techniques (AgST), such as dot–blot (DB) or ELISA, among others. Currently, there are a wide variety of available techniques and commercial kits that allow us to identify the presence of AILD-related autoantibodies. Unfortunately, the sensitivity and specificity of these tests may vary and, in most cases, the performance of these techniques varies depending on the commercial brand employed. Moreover, the interpretation of some of these results requires trained personnel [19]. In addition, the lack of a common working algorithm for the detection of AILD-related autoantibodies results in a substantial degree of variability in the management of autoantibodies detection among the different laboratories.

In this context, different quality assurance programs for diagnostic serology in liver disease are available [20]. In Spain, GECLID autoimmunity program number 1 (https://www.geclid.es/course/category.php?id=55 (accessed on 20 December 2021)) is framed within the Spanish Immunology Society. Every two months, two different samples are sent to participating laboratories to systematically assess the autoantibodies testing performance of each participating laboratory. However, challenging samples are still a matter of concern in daily practice. For this reason, given the central role of autoantibodies in the diagnosis of AILD and the wide number of methods that are available for their detection, the Spanish Autoimmunity Group of the Spanish Immunology Society (GEAI-SEI) (https://www.inmunologia.org/index.php/grupos-sei/autoinmunidad-geai (accessed on 5 December 2019)) carried out a thorough intercomparison study of challenging samples in 2020 (Autoimmunity Workshop GEAI-SEI 2020).

**Table 1 diagnostics-12-00697-t001:** Clinical significance of autoantibodies detected in AIH and PBC.

	Autoantibodies	Sensitivity	Specificity	Clinical Features
**Primary Biliary Cholangitis (PBC)**	AMA-M2	84.5% [21]	97.8% [21]	No difference in clinical features between AMA-positive and AMA-negative patients.
	PDC-E2	80–90% [22,23]		
	OGDC-E2	20–60% [22,23]		
	BCOADC-E2	50–80% [22,23]		
	Sp100	21.3–24.9% [9]	97.3–98.1% [9]	Severe disease and clinical outcome. Worse outcome and likely a more rapid progression of PBC. Marker of poor prognosis [9].
	gp210	25.7–28.8% [9]	98.2–98.8% [9]	Severe disease and clinical outcome. Hepatic failure-type progression [23]. Advanced stages of disease and faster disease progression rates. Marker of poor prognosis [9].
**Autoimmune Hepatitis (AIH)**	ANA	23–83% [24]	69–94% [24]	Diagnostic of AIH-1. Not associated with disease course or outcome.
	SLA	32.6% type 1 [25]	100% [25]	Diagnostic of AIH-1. More severe disease and a worse outcome, propensity for relapse after corticosteroid withdrawal [26].
	LKM-1	57% type 2 [27]	100% [28]	Diagnostic of AIH-2 particularly in the absence of hepatitis C virus infection. Associated with younger age at presentation, more frequently associated with fulminant hepatic failure and with partial IgA deficiency [29].
	LC-1	35% type 2 [30]	98% [30]	Diagnostic of AIH-2 particularly in the absence of hepatitis C virus infection. Liver inflammation and rapid progression to cirrhosis [30].
	SMA/F-Actin	23–52% type 1 [24]	93–100% [24]	SMA: Diagnostic of AIH type-1. Not associated with disease course or outcome; SMA/F-Actin: Associated with younger age at disease onset, treatment dependence in children. Predicts progression to liver failure and need for transplantation [31].

Abbreviations: AMA-M2, anti-mitochondrial autoantibodies type-2; ANA, anti-nuclear autoantibodies; BCOADC-E2, E2-subunit of branched chain 2-oxo-acid dehydrogenase complex; gp210, glycoprotein 210; LC-1, liver cytosol-1; LKM-1, liver-kidney microsomal type-1; OGDC-E2, E2-subunit of oxoglutarate dehydrogenase complex; PDC-E2, E2-subunit of pyruvate dehydrogenase complex; SLA, soluble liver antigen; SMA/F-actin, smooth muscle autoantibodies directed to F-actin. AIH-1, autoimmune hepatitis type 1; AIH-2, autoimmune hepatitis type 2.

The main objectives of this nationwide study were (i) to compare the working algorithms of the 40 participating laboratories and (ii) to analyse the results related to AIH- and PBC-associated autoantibodies obtained from the samples of 17 patients that were distributed to the 40 laboratories and tested using different methods. Based on the obtained results, some recommendations to improve the diagnostic accuracy of in vitro immunological tests and the effectiveness of working algorithms are suggested.

## 2. Materials and Methods

### 2.1. Autoimmunity Workshop GEAI-SEI 2020 Design

A total of 40 laboratories of different Spanish clinical institutions voluntarily participated in this multicentre study organised by GEAI-SEI (Figure 2). The immunology departments of Hospital Clínic de Barcelona and Hospital Universitari Son Espases selected 17 patients with liver diseases that attended the hepatology or gastroenterology departments of either hospitals. These 17 patients were selected according to the already-known presence of different autoantibodies and complex medical history related to AILD. Unfortunately, no LKM-1 patient was included. Serum samples from these 17 patients were sent to participating laboratories (*n* = 40) together with brief summaries of the patients’ medical histories, but without autoantibody data. One laboratory, which enrolled after the deadline, received only eight serum samples. Therefore, each patient’s sample was analysed 39–40 times by the different laboratories, raising the number of total complete patient reports to almost 680. Each laboratory decided, based on its own daily working algorithm and the clinical data given, how to carry out the study in terms of the different techniques performed to determine AILD-associated autoantibodies for each patient (Appendix A). All results of the present Autoimmunity Workshop GEAI-SEI 2020 were reported anonymously, in order to preserve the confidential nature.

### 2.2. Patients

The diagnoses of the 17 patients included in this study were: PBC (*n* = 6), AIH (*n* = 3), PBC/AIH-1 overlap (*n* = 1), AIH/Toxic (*n* = 1), pre-PBC (*n* = 3), PSC (*n* = 1), hepatitis C virus chronic liver disease (*n* = 1) and dissociated cholestasis with hepatomegaly (*n* = 1). Liver biopsy data for each patient were available, except for those classified as pre-PBC and dissociated cholestasis with hepatomegaly. Clinical, serological and histopathological data were collected from patient’s medical histories (Appendix A). This study was conducted in accordance with the Declaration of Helsinki and was approved by the Research Ethics Committee of the Hospital Clínic de Barcelona (HCB/2019/0808). Written informed consent was obtained from all patients.

### 2.3. Methods

Assays were classified as: (a) indirect immunofluorescence assays (IFA) or (b) AgST. Commercial assays were performed following the manufacturers’ instructions.

(a)
*Indirect immunofluorescence assays:*


IFA was used for AILD-associated autoantibodies detection by all laboratories, as described by Detrick et al. [32]. As is well known, complex multiantigenic substrates (tissues and cells) are used for IFA in autoimmunity. Slides for IFA-RTT and/or for IFA-HEp-2 from different manufacturers were used by each laboratory. Both titre and pattern data were collected. Results for ANA, anti-SMA/F-actin, anti-LKM-1, anti-LC-1, AMA-M2 and anti-gastric parietal cell (GPC) autoantibodies were reported by IFA-RTT. Through IFA-HEp-2, ANA and anti-cytoplasmic autoantibodies were reported, according to ICAP nomenclature [33]. Moreover, studies by IFA on human neutrophils (IFA-hN) were also performed by some laboratories, as described by Detrick et al. [34].

The most commonly used sera screening dilutions were 1/40 (52.5%), 1/160 (57.5%) and 1/20 (69.4%) for IFA-RTT, IFA-HEp-2 and IFA-hN studies, respectively. More than 75% of laboratories employed an anti-IgG as secondary antibody for each IFA technique and >50% of laboratories revised IFA slides directly on a microscope (Appendix A).

(b)
*Antigen-specific techniques (AgST):*


In contrast to IFA, isolated antigens are used in AgST. Different types of AgSTs, including IFA on VSM47 line cell, DB, ELISA, chemiluminescence-immuno-assay, fluoro-enzyme-immunoassay from different manufacturers were employed depending on the laboratory, as described by Detrick et al. [35]. The AgST to be performed was decided based on the patients’ clinical history, the working algorithm and/or their own IFA results. Thus, the number and type of results to be compared varied for each autoantibody. In relation to AMA-M2, different detection methods containing either a mixture of recombinant human E2-subunits, a fusion protein based on the three E2-subunits (M2-3E2), a mixture of native and recombinant antigens or the three isolated recombinant E2-subunits were used. Most of the participating laboratories in this workshop (>75%) used DB as the AgST of choice (Appendix A).

### 2.4. Statistical Methods

Consensus outcomes for every result category from each patient were established with a minimum of 75% concordance between the results obtained by all laboratories, according to GECLID AU-prospectus document (https://www.geclid.es/mod/resource/view.php?id=4497 (accessed on 5 June 2020)).

## 3. Results

### 3.1. Working Algorithms

The working algorithms employed by participating laboratories (*n* = 40) for immunological testing of AILD showed a wide range of different protocols (Appendix A). The most frequent algorithm was reported by 11/40 (27.5%) participants and included an initial IFA-RTT and IFA-Hep-2 screening followed by confirmation of positive corresponding patterns by AgST, mainly by DB. In cases of high clinical suspicion of AILD, although IFA findings were negative, these laboratories performed a DB as extended screening. The second predominant option, followed by 9/40 (22.5%) laboratories, also initiated the study by IFA-RTT and IFA-HEp-2 but only performed AgST when IFA tests were positive. ANCA were added to IFA screening by 7/40 (17.5%) laboratories. One third (6%) and a quarter (4%) of them included DB or IgA anti-tissue transglutaminase in the extended study, respectively. A varied initial screening adding IgA anti-tissue transglutaminase, HLA typing or DB to IFA was performed by 6/40 (15%) participants. Moreover, the study of ANCA and immunoglobulin levels was added in some occasions. Another option was to use different algorithms according to diagnostic suspicion. Indeed, 3/40 (7.5%) laboratories reported IFA-RTT and IFA-HEp-2 results if PBC was the clinical suspicion and added DB and ANCA only if AIH was the expected diagnosis. Two participants (5%) used IFA-RTT and/or IFA-HEp-2 depending on the clinician’s request. One participant (2.5%) performed DB as an autoantibody detection method in initial AILD screening and IFA was employed only to confirm positive results. Finally, one laboratory (2.5%) performed IFA-RTT, but not IFA-HEp-2, as the first screening, but simultaneously used DB in all sera, except in patients positive for hepatitis C virus.

### 3.2. Autoimmune Hepatitis

AIH is a chronic inflammatory liver disease that affects almost all age groups worldwide. Although the aetiology of this disease is unknown, AIH is considered as an autoimmune T-cell mediated disease resulting from the disruption of immune tolerance against liver tissues promoted by the impairment of liver-antigen-specific regulatory T cells (i.e., CD4^+^ CD25^high^FOXP3^+^) [36]. The mean annual incidence of the disorder in Caucasian Northern Europeans ranges from 1.07 to 1.90 per 100,000 individuals, with a point prevalence of 16.9 per 100,000 [37]. AIH predominates in females, but it can affect children or adults of both genders [37]. The diagnosis of AIH is based on histological abnormalities (interface hepatitis), characteristic clinical, and laboratory findings (elevated serum aspartate aminotransferase and alanine aminotransferase levels and increased serum IgG concentration), and the presence of one or more characteristic autoantibodies [38,39]. The presence of other causes of liver disease, such as viral hepatitis, must be excluded.

The detection of AIH-related autoantibodies is very useful in the diagnosis of AIH. Several autoantibodies have been associated with a different clinical presentation and prognosis. These associations have led to the classification of AIH-1 and AIH-2 [2]. Autoantibodies associated with AIH-1 are ANA, anti-SMA and anti-SLA. AIH-2 (about 5–10% of all AIH patients) is typically defined by anti-LKM-1 or, in rare cases, by anti-LKM-3 and/or anti-LC1 [30,40].

Among the 17 patients included in our study, three (#4, #7 and #10) were diagnosed with AIH, one more patient (#5) had a differential diagnosis with toxic hepatitis (AIH/Toxic). Additionally, one patient (#8) was diagnosed as an overlap syndrome PBC/AIH-1. All of them have data from liver biopsies (Appendix A).

#### 3.2.1. Anti-Nuclear Autoantibodies (ANA)

Background

ANA were the first autoantibodies detected in AIH patients more than 50 years ago [41]. These autoantibodies are targeted against different molecules that are found in the cellular nucleus, such as centromeres, histones, double-stranded DNA, chromatin, or ribonucleoprotein complexes, among others. ANA are currently the most sensitive marker of AIH, frequently observed as a speckled or homogenous pattern in IFA-HEp-2 (AC-1, AC-4, AC-5 according to ICAP). However, this technique is not specific for AIH given that ANA can also be detected in healthy people, patients with other autoimmune diseases or patients with other liver diseases, such as fatty liver disease, drug-induced liver injury (DILI) disease, or viral hepatitis [2].

ANA is detectable as nuclear staining in IFA-HEp-2 and IFA-RTT [42]. However, a much clearer definition of ANA pattern is obtained using HEp-2 cells. The use of HEp-2 as a screening method for ANAs has been traditionally questioned given the high rate of healthy subjects with a positive result for ANAs when tested at low dilutions (1:40).

Results

All laboratories (40/40; 100%) used IFA-HEp-2 but only 20/40 (50%) used IFA-RTT for the detection of ANA (Figure 3A). Concerning patients #4 and #7 (AIH-1), 78/80 (97.5%) of ANA results by IFA-HEp-2 were positive. However, when analysing the sera of these two patients by IFA-RTT, 32/40 (80%) tests were reported as positive. Additionally, patient #7 was tested for CENP-B (*n* = 14) and CENP-A (*n* = 7) by AgST, with all the results positive.

The remaining three AIH-diagnosed patients (#5, #8 and #10) were classified as ANA negative by consensus (84.2%, 84.2% and 94.7%, respectively) using IFA-Hep-2. However, 26/120 (21%) IFA-Hep-2 tests were reported as positive at different dilutions, ranging from 1:80 to 1:1280.

Conclusions

Our results confirm that the detection of ANA by IFA-Hep-2 is more appropriate owing to the higher sensitivity of this method. Indeed, all laboratories studied ANA using IFA-Hep-2 and only 50% of participants reported ANA by IFA-RTT. Reporting only the presence or absence of ANA when tested by IFA-RTT is highly recommended, given that a precise ANA pattern is not required in the clinical context of AIH added to the low specificity of ANA at low titres. Furthermore, considering that recognising the presence of ANA by IFA-RTT could be challenging, the evaluation of ANA on Hep-2 cells is a recommended alternative to rat tissue sections. Moreover, the presence of low-titre ANA could be easily solved by increasing the screening dilution to 1/160 titre as recommended by international groups [43,44].

#### 3.2.2. Anti-Smooth Muscle/F-Actin (SMA/F-Actin) Autoantibodies

Background

Anti-SMA were first described by Johnson et al., in 1965 [45]. They belong to a heterogeneous group of autoantibodies that react with protein subunits of different types of cytoplasmic filaments (microfilaments, microtubules or intermediate filaments) [46]. Anti-SMA are present in about 50% of patients with AIH-1 and can be the only detectable autoantibody [2].

For the study of anti-SMA/F-Actin by IFA-RTT, Botazzo and colleagues [47] distinguished three different patterns: SMA-V (vessels), SMA-VG (vessels, glomeruli) and SMA-VGT (vessels, glomeruli, tubuli). Anti-SMA/F-Actin is characterised by the staining of contractile fibrils around renal tubules (dashed line) (Figure 1A). SMA showing reactivity against F-actin are more specific for AIH, but can also be detected in other liver diseases [48]. Regarding IFA-Hep-2, only AC-15 fibrillar pattern is specifically related to the presence of anti-SMA/F-Actin but there are two additional cytoplasmic fibrillar patterns (AC-16, AC-17) which, depending on the substrate employed, are not easy to differentiate from AC-15, even for expert observers.

Results

From the 17 analysed patients, three of them (#1, #7 and #8) were classified as positive by consensus for anti-SMA/F-Actin according to the established classification described above for IFA-RTT. Almost all laboratories (99.2%) performed IFA-RTT and IFA-HEp-2 (Table 2).

In relation to IFA-RTT, even though the presence of anti-SMA was identified in 103/119 (86.6%) of the results, only 18/103 (17.4%) were reported as positive for anti-SMA/F-Actin. Regarding IFA-HEp-2, only 26/117 (22%) results were reported as fibrillar compatible with anti-SMA. From these 26 fibrillar positive results, 17/26 (65%) were specifically reported as AC-15. AgST was performed in 71/119 (59%) samples related to SMA/F-actin positive patients (#7, #1 and #8) being 48/71 (67.6%) positive for anti-SMA/F-actin.

Conclusions

The pattern of SMA-VGT is the most specific for anti-SMA/F-actin autoantibodies. For this reason, the correct detection or identification of this IFA pattern arises as an important issue. In our cohort, most laboratories were able to identify the presence of anti-SMA. However, only a few of them reported the presence of the SMA/F-actin compatible pattern on IFA-RTT in this study. Interestingly, some of the participants also detected a compatible cytoplasmic pattern on HEp-2 cells (AC-15). Concerning AgST, the corresponding results are independent of the expertise of the observer, with the exception of IFA-VSM47 method that shows a great variability between different laboratories. Overall, AgSTs were able to detect 67% of positive samples. For this reason, it was found that diagnostic sensitivity increased when both techniques (IFA and AgST) were performed simultaneously. The results obtained also highlight the importance of training laboratory professionals for the detection of IFA patterns.

#### 3.2.3. Anti-Soluble Liver Antigen/Liver Pancreas (SLA) Autoantibodies

Background

Among all AIH-1-associated autoantibodies, anti-SLA shows the highest specificity [42]. This autoantibody targets the antigen O-phosphoseryl-tRNA:selenocysteine-tRNA synthase, a synthase that converts O-phosphoseryl-tRNA to selenocysteinyl-tRNA [25]. Anti-SLA/LP autoantibodies are present in about 10–30% of cases of AIH, occasionally found in patients with AIH who are negative for ANA, anti-SMA and anti-LKM-1 autoantibodies. Anti-SLA/LP autoantibodies are also detectable in subgroups of paediatric patients with autoimmune cholangitis or in adult patients with HCV infection [49]. Anti-SLA autoantibodies cannot be detected using IFA techniques; they can only be detected by AgST.

Results

Two patients (#4 and #7) reached a positive consensus for anti-SLA autoantibodies. Hence, we analysed 80 reports regarding the two positive samples for anti-SLA autoantibodies. We found that most of the reported results 72/80 (90%) were performed using AgST (DB), in addition to IFA-RTT, being 66/72 (91.7%) positive, 4/72 (5.6%) negative and 2/72 (2.8%) inconclusive for SLA (Figure 3A). On the contrary, 8/80 (10%) were analysed exclusively by IFA-RTT leading to a false negative result in all of them.

Among the patients classified as negative for anti-SLA autoantibodies (*n* = 15), from 548 results reported by AgST for SLA, none of these results were positive. Thus, the specificity of AgST for the detection of anti-SLA autoantibodies is 100% in our cohort.

Conclusions

Few participating laboratories did not include AgST when IFA tests were negative (4/40; 10%). This is particularly inadequate in high suspicion of AIH cases because anti-SLA autoantibodies cannot be detected by IFA. Therefore, AgST for the detection of anti-SLA autoantibodies should be performed independently of the result obtained by IFA in case of clinical suspicion of AIH.

#### 3.2.4. Anti-Liver Cytosol Type-1 (LC1) Autoantibodies

Background

The enzyme formiminotransferase cyclodeaminase has been identified as the molecular target of anti-LC-1 autoantibodies [50]. The detection of anti-LC-1 autoantibodies alone or in association with anti-LKM-1 strongly suggests the diagnosis of AIH-2. However, they are not specific to AIH and can be detected in HCV [51]. It is important to underline that the pattern of anti-LC-1 autoantibodies (a brightly staining of the hepatocyte cytoplasm, sparing the centrilobular zone) by IFA-RTT can be masked when they appear in combination with anti-LKM-1 autoantibodies. In this case, AgST help to identify anti-LC-1 autoantibodies.

Results

One patient (#10) diagnosed as AIH-2 was classified as positive by AgST for anti-LC-1 autoantibodies according to the established consensus.

All laboratories performed IFA-RTT when analysing the sample of this patient. Surprisingly, only 11/40 (27.5%) of the participants specifically detected the presence of anti-LC-1 autoantibodies. Additionally, 5/40 (12.5%) reported a pattern partially compatible with the presence of anti-LKM-1 autoantibodies and 9/40 (22.5%) reported a positive result but without a pattern (Figure 3A). Finally, 15/40 (37.5%) reported a negative result by IFA-RTT.

In relation to IFA-HEp-2 results, it should be noted that 25/40 (62.5%) of the results reported an anti-cytoplasmic autoantibodies pattern corresponding to a reticular pattern (AC-21), suggesting the presence of AMA.

Independently of the results obtained by IFA-RTT, all laboratories performed AgST for LC-1 autoantibody detection. In this case, 40/40 (100%) of the results obtained by AgST (DB) were positive for anti-LC-1 and negative for anti-LKM-1 autoantibodies. Among the 581 results derived from 16 patients classified as negative for anti-LC-1 autoantibodies, all of them were negative except one that was reported as “inconclusive” when tested by AgST (DB). Thus, the specificity of AgST for anti-LC-1 autoantibodies is almost 100% in our study.

Conclusions

A frequent problem in the interpretation of IFA results occurs when the observed staining does not clearly correspond to the expected pattern related to liver disease-associated autoantibodies. A distinct example of this problem is in IFA results corresponding to patient #10, where in addition to liver staining consistent with the presence of anti-LC-1 autoantibodies, a kidney staining was also observed. Only 27.5% of the participants detected the presence of anti-LC-1 autoantibodies by IFA-TTR. Taking into account the difficulty of the participating laboratories to detect the presence of anti-LC-1 autoantibodies by IFA-TTR, it is recommended to use AgST in case of suspicion of AIH-2.

#### 3.2.5. Anti-Liver Kidney Microsomal-1 Autoantibodies

Despite the absence of positive patients for anti-LKM-1 autoantibodies in this study, an important limitation, there were two patients (#5 with a diagnosis of AIH/Toxic and #10 with a diagnosis of AIH-2) who presented an IFA-RTT pattern that could be compatible with the presence of these autoantibodies. The presence of anti-LKM-1 autoantibodies was discarded by AgST in our patients.

### 3.3. Primary Biliary Cholangitis

PBC is an autoimmune disease characterised by the progressive destruction of the intrahepatic bile canaliculi. This progressive destruction generates liver damage and cholestasis of variable severity. The interaction of chronic immune damage with biliary epithelial cells loss is considered the main cause of the disease, although this mechanism is still under study. The prevalence is low, but it predominantly affects women in their fifth decade of life. It is often asymptomatic and patients have variable risks of progressive ductopenia, cholestasis, and biliary fibrosis [52]. Diagnosis is usually based on the presence of serum liver tests indicative of cholestatic hepatitis in association with circulating antimitochondrial antibodies [4], specifically anti-mitochondrial autoantibodies (AMA) type-2 (AMA-M2). They recognise the E2-subunits of the 2-oxo-acid dehydrogenase complex (pyruvate dehydrogenase complex (PDC-E2), branched-chain 2-oxo-acid dehydrogenase complex (BCOADC-E2) and 2-oxo-glutarate dehydrogenase complex (OGDC-E2)) and/or other PDC proteins: E1α subunit and E3 binding protein [22,23,53,54]. As mentioned above, PBC-specific nuclear antigens have been identified, being Sp100 and gp210 the major target antigens. New serological markers recently associated with PBC (anti-hexokinase-1 and anti-kelch-like-protein-12 autoantibodies [10,11]) have not been studied in this workshop due to the fact that currently there are no available CE-marked kits on the market.

Among the 17 patients included in our study, 6 had been diagnosed as PBC (#6, #11, #12, #13, #15 and #16), one as PBC/AIH-1 overlap (#8), and three as pre-PBC (#3, #9 and #17) (Appendix A).

#### 3.3.1. Anti-gp210 Autoantibodies

Background

Gp210 is a type I integral membrane protein that anchors nuclear pore complexes to the nuclear membrane. ANA against gp210 is present in approximately 20% of PBC patients, and in around 30–50% of AMA-negative patients [2,22]. In fact, these autoantibodies correlate with PBC disease stage and have been associated with a worse outcome and rapid progression. According to ICAP, gp210 pattern (AC-12) is characterized by punctate staining of the nuclear envelope in interphase cells, with accentuation of fluorescence at the points where adjacent cells touch each other. No staining of the metaphase and anaphase chromatin plates is observed. This pattern is also associated with other antigens, including p62 nucleoporin, LBR, and Tpr; but specific immunoassays for these autoantibodies are not currently available as commercial tests [55,56,57,58].

Results

From the 17 patients, three of them (#3, #16 and #17) showed AgST positive results confirming the presence of autoantibodies against gp210 by consensus agreement.

Only 61/119 (51.3%) of the potential results were reported by IFA-RTT, indicating that almost half of the expected results were not performed. Surprisingly, 35/61 (57.4%) of these results were compatible with a pattern associated with the presence of anti-gp210 autoantibodies. Among the rest of the results, 10/61 (16.4%) reported a positive ANA but without specifying a pattern, 6/61 (9.8%) reported an incorrect pattern and a negative result was indicated by 10/61 (16.4%) laboratories (Figure 3B).

The study of samples by IFA-HEp-2 was performed in all laboratories for these three PBC patients. Only 73/119 (61.3%) of laboratories indicated a pattern (AC-12) or combination of nuclear membrane patterns (AC-11/AC-12), associated with the presence of anti-gp210 autoantibodies by IFA-HEp-2.

Regarding AgST, 105/110 (95.5%) were positive for anti-gp210 autoantibodies. Moreover, among the patients classified as negative by consensual agreement for anti-gp210 autoantibodies (*n* = 14), we received 512 results obtained by DB, being 510/512 (99.6%) negative. Thus, AgST had high specificity in our cohort.

Conclusions

The interpretation of fluorescence patterns by IFA-RTT and/or IFA-HEp-2 can be challenging for anti-gp210 autoantibodies, and could lead to false negative results in routine practice. The comparison between IFA and AgST clinical performances showed relatively low concordance, being the sensitivity of AgST clearly greater than IFA (95.5 vs. <65%) for anti-gp210 autoantibodies. Therefore, to confirm the presence of anti-gp210 autoantibodies, an antigen-specific test is highly recommended.

#### 3.3.2. Anti-sp100 and Promyelocytic Leukaemia Protein (PML) Autoantibodies

Background

Sp100 is a nuclear protein with an unknown function, co-localised in nuclear dots with PML, which is a transformation and cell growth suppressing protein expressed in promyelocytic leukemia cells [59]. As PML co-localises with Sp100, the pattern of ANA by IFA is indistinguishable from the pattern associated with anti-Sp100 autoantibodies and rarely occurs in the absence of anti-Sp100 autoantibodies. Therefore, only AgST allowed us to confirm or rule out the presence of these autoantibodies. Anti-Sp100 ANA are present in around 30–50% of AMA-negative PBC patients [2,22] although they could be also detected in some other conditions (systemic autoimmune diseases and hepatotropic virus infections) [60]. In PBC patients, these autoantibodies are a marker of poor prognosis as they are associated with severe disease and clinical outcome with a more rapid progression of the disease [9]. According to ICAP, Sp100 pattern (AC-6) is characterised by countable discrete nuclear speckles (6 to 20 nuclear dots/cell).

Results

Three of our studied patients (#9, #13 and #15) showed the presence of autoantibodies against Sp100 in their sera by agreement, employing AgST.

Only 53/117 (45.3%) of potential results were reported by IFA-RTT. From these, only 13/53 (24.5%) corresponded to a pattern compatible with the presence of anti-Sp100 autoantibodies. In contrast, 116/117 (99.1%) results were reported using IFA-HEp-2, being 99/116 (85.4%) positive for a pattern compatible with the presence of anti-Sp100 autoantibodies (AC-6/AC-7). Additionally, 13/116 (11.2%) reported a positive result but with a different ANA pattern by HEp-2.

In regards to AgST, 113/117 (96.6%) of the results were positive for anti-Sp100 autoantibodies. Among the patients classified as negative for anti-Sp100 autoantibodies (*n* = 14), we received 476 results obtained by DB. This AgST had high specificity in our cohort since 472/476 (99.2%) of the results were negative. From the three patients mentioned above, only one (#9) reached a positive consensus agreement (26/27; 96.4%) for anti-PML autoantibodies.

Conclusions

IFA-HEp-2 and AgST showed high sensitivity and specificity to detect ANA against Sp100, although AgST sensibility/sensitivity was slightly greater. However, IFA-RTT studies can be challenging and could lead to false negative results in clinical practice.

#### 3.3.3. Anti-Mitochondrial (AMA) Autoantibodies

Background

The presence of AMA-M2 is the hallmark of PBC. AMA-M2 recognise the E2-subunits of the 2-oxo-acid dehydrogenase complex (pyruvate dehydrogenase complex (PDC-E2), branched-chain 2-oxo-acid dehydrogenase complex (BCOADC-E2), and 2-oxo-glutarate dehydrogenase complex (OGDC-E2)) and/or other PDC proteins: E1α subunit and E3 binding protein [22,23,53,54].

On IFA-RTT, AMA-M2 autoantibodies show a characteristic pattern, characterised by cytoplasmic granular fluorescence in (i) liver hepatocytes, (ii) distal and proximal (P) tubules of the kidney (Distal > P1-P2 > P3, in terms of fluorescence intensity) and (iii) stomach GPC. In IFA-HEp-2, according to ICAP, AMA pattern (AC-21) is associated with a coarse granular filamentous staining extending throughout the cytoplasm. However, several subtypes of AMA (traditionally named M1-M9) have been described in IFA-RTT. Among AMA positive patients, AMA-M2 are found in 95–98% of PBC patients. AMA against M4, M8 and M9 are also associated with PBC, but with a poor prevalence [2]. In spite of this clinical relevance, AMA subtypes cannot be distinguished by IFA-HEp-2.

Diverse AgST commercial kits for the detection of AMA-M2 in clinical samples are currently available. Different AMA-M2 detection methods containing either a mixture of recombinant human E2-subunits, a fusion protein M2-3E2, a mixture of native and recombinant antigens or the three isolated recombinant E2-subunits are commonly used in the daily routine by different clinical institutions. In our cohort, patients with PBC or pre-PBC diagnostic were studied by different methods containing different antigens. These results are listed in Table 3.

Results

Considering only PBC (patients confirmed by biopsy), AMAs were positive by substantial agreement in 5/7 (#11, #12, #13, #15 and #16) employing different AgST and IFA-RTT. However, less than 20% of laboratories detailed whether the detected AMA pattern in these patients with PBC by IFA-RTT was AMA-M2 or another subtype, indicating only the result of “AMA positive”.

All those PBC patients with a positive result by IFA-RTT also showed a cytoplasmic pattern compatible with the presence of AMA (AC-21) by IFA-HEp-2, being 4/5 (80%) positive by consensus and 1/5 (20%) nearly to positive consensus. A negative consensus was achieved in the rest of the patients using IFA-RTT and IFA-HEp-2, except for one patient (#8). In this case, only 55% of the laboratories reported a negative result by IFA-HEp-2.

Those patients with negative results by IFA were also negative by some AgST. However, we found positive results by consensus using methods based on the detection of the M2-3E2. Interestingly, the presence of AMA directed exclusively against BCOADC-E2 reached a substantial agreement in one patient (#8). In the case of patient #6, 71% (20/28) of positive results were obtained by one kind of DB for anti-M2-3E2, which could not be reproduced by other methods. In IFA-RTT positive patients, using AgST based on the recognition of the three subunits separately, we detected AMA-M2 directed against PDC-E2 in 5/5 and BCOADC-E2 in 2/5 (Table 3).

Regarding the patients classified as pre-PBC (*n* = 3), 2/3 (#3 and #17) were negative for AMA by IFA-TTR, but only one of them by substantial agreement (#17). The other patient (#9) was classified as positive but without agreement. In this case, only 26/39 (66.7%) laboratories indicated a positive result by IFA-TTR. In relation to IFA-HEp-2, all 3 pre-PBC patients were considered negative being 2/3 (66.7%) by substantial agreement.

Although no consensus was reached on the presence of AMA using IFA, a positive agreement was obtained in patient #9 using AgST for BCOADC-E2 (6/6). For patient #17, 3/5 (60%) laboratories that tested AMA-M2 against native AMA gave positive results (Table 3).

The presence or absence of anti-GPC autoantibodies is usually reported in IFA-RTT studies. However, the detection of these autoantibodies is challenging in the presence of AMA-M2 because the characteristic staining pattern of AMA-M2 also stains GPCs. Surprisingly, in those patients with positive consensus for the presence of AMA, 146/192 (76%) of them also showed a consensus for the absence of anti-GPC autoantibodies by IFA-RTT even though AgST for autoantibodies to gastric ATPase were only performed in 18/386 (4.7%) cases (Table 3).

Conclusions

Although AgSTs remain the gold standard for AMA-M2 detection, IFA-RTT and IFA-HEp-2 can provide valuable information about these autoantibodies. The main differences for AMA-M2 detection were due to patients who lacked autoantibodies against the dominant epitope of 2-oxo-acid dehydrogenase complex, PDC-E2. Although those methods containing the three E2-subunits separately were the most sensitive for detecting the positivity of anti-BCOADC-E2 autoantibodies, few laboratories routinely use them. Some reactivity was also detected against OGDC-E2 (patient #17) although no consensus was reached. Nonetheless, it is worth noting the good sensitivity of methods based on the M2-3E2 to detect patients with AMA directed against BCOADC-E2 (patient #8), even in patients with negative results by IFA-RTT. Nevertheless, there are some AMA-M2 that only recognise E1α or E3 binding protein from the PDC and the addition of native purified PDC is highly recommended for AMA-M2 testing [53].

As for the anti-GPC autoantibodies study, is highly recommended to use AgST for the detection of anti-gastric ATPase autoantibodies when a compatible AMA-M2 pattern is detected by IFA-RTT. If an AgST is not available for its evaluation, the most correct attitude in this situation would be to advise clinicians that anti-GPC autoantibodies are not assessable in AMA-M2 positive samples.

### 3.4. Other Autoantibodies

#### 3.4.1. Anti-Neutrophil Cytoplasmic Autoantibodies

Only a variable subset of participants tested each serum for ANCA. For this reason, although ANCA positivity was found in 5/17 (29.4%) patients (#4, #7, #8, #14 and #17), a consensus was reached only in one patient (Figure 4). The diagnosis of this patient (#14) was PSC. This sample was evaluated by 32/39 (82.1%) laboratories and a positive result was given by 29/32 (90.6%). The majority of the results corresponded to aANCA (22/32; 68.8%) and 6/32 (18.8%) laboratories give a perinuclear pattern (pANCA). An additional laboratory did not differentiate between aANCA or pANCA (3.1%). Those patients (4/17; 23.5%) that were found positive for ANCA by some laboratories but without consensus were diagnosed as AIH-1 (#4 and #7), PBC/AIH overlap (#8) and pre-PBC (#17). Anti-mieloperoxidase and anti-proteinase-3 autoantibodies were reported negative in all cases.

#### 3.4.2. Anti-Ro52 Autoantibodies

Among the 17 patients included, four of them (#1, #2, #5 and #17) showed results that indicated the presence of autoantibodies against Ro52 employing AgST.

From the four positive patients (#1, #2, #5 and #17) consensus was reached by three of them (#1: 69.2%, #2: 82.9%, #5: 88.2% and #17: 91.4%). Different AgST currently exist to analyse anti-Ro52 autoantibodies, but most of the laboratories (27/40; 67.5%) used the same DB. Moreover, each laboratory tested a variable proportion of patients (from 1 to 17 samples).

For the four positive samples, 105/107 (98.1%) DB results were positive while only 24/44 (54.5%) non-DB results were positive. From the 13 negative samples, there were 353 DB results and 88 non-DB results and only 6/441 (1.4%) positive results were reported (2 doubtful and 1 positive by DB and 3 positives by non-DB AgST).

#### 3.4.3. Anti-Centromere Autoantibodies

Although anti-centromere autoantibodies are diagnostic markers of systemic sclerosis, they are also detectable in 10–30% of patients with PBC without any apparent clinical manifestations of concomitant systemic sclerosis [22].

Positive consensus for anti-centromere autoantibodies was obtained for 3 out of all 17 patients included in the study (#1, #3 and #7): 95–100% by IFA-HEp-2 (AC-3) and 100% by anti-CENP-A and CENP-B AgST. Diagnoses of liver disease for these patients (#1, #3 and #7) were Hepatitis C Virus Chronic Liver Disease, pre-PBC and AIH1, respectively. Furthermore, all 3 patients exhibited some signs of systemic sclerosis: patient #1 was diagnosed with limited systemic sclerosis, patient #3 suffered pre-scleroderma and patient #7 presented with Raynaud’s phenomenon.

In none of the other patients diagnosed with PBC (*n* = 6) or pre-PBC (*n* = 2) was detected a pattern compatible with the presence of these autoantibodies by IFA-Hep-2 (AC-3).

## 4. Discussion

The presence of autoantibodies plays a central role in the diagnosis and classification of AILD. However, there are no consensus guidelines on working algorithms, as well as on the most appropriate methods, that should be used to detect these autoantibodies. This fact causes significant differences among autoimmunity laboratories. This study aimed for the first time to compare the working algorithms and the methods used by an important number of autoimmunity laboratories (*n* = 40) nationwide in the context of AILD diagnosis.

Regarding the working algorithms, most of the laboratories performed an initial screening by IFA-RTT and IFA-HEp-2 and upon request or clinical suspicion extend this screening by AgST, mainly DB. About 40% of participants did not include AgST when IFA results were negative. This is particularly inadequate in high suspicion AIH because SLA autoantibodies cannot be detected by IFA. In the present study, the sensitivity of AgST was greater than IFA for almost all the autoantibodies (Figure 3). For example, IFA showed a poor sensitivity for some PBC specific autoantibodies, such as anti-gp210 (Figure 3B). ANCA are included in the initial or extended protocol only in a minority of laboratories (Appendix A) and in some cases only when there is suspicion of PSC. In addition, the number of laboratories that determine immunoglobulins, IgA anti-tissue transglutaminase and HLA typing is even lower. Finally, only one laboratory performs the complete protocol as suggested and published in 2018 by the GEAI-SEI [61]. In conclusion, a wide range of working protocols is used by Spanish clinical laboratories for AILD immunological diagnosis.

We also analysed the results of AIH and PBC-associated autoantibodies obtained by the 40 laboratories participating in the Workshop GEAI-SEI 2020 using different methods (including IFA or AgST) for those patients included in this work (*n* = 17). Following that, some recommendations regarding the most appropriate techniques have been generated.

In reference to the immunological diagnosis of AIH, it is important to correctly report the pattern of anti-smooth muscle/F-Actin autoantibodies on RTT, taking Bottazzo’s classification into account. Moreover, it is highly recommended to carry out AgSTs complementary to the IFA-RTT studies, regardless of the result obtained in the IFA-RTT (positive or negative). AgSTs have good sensitivity and specificity for all the mentioned autoantibodies and are not as dependent on the experience of the observer as IFA-RTT. It is especially important when (i) AIH diagnosis is suspected and IFA-RTT study reveals a negative result because anti-SLA autoantibodies are not detectable by IFA-RTT and the only way to identify them is by using AgSTs; (ii) when the presence of an anti-LKM-1 autoantibody may mask the presence of an anti-LC-1 autoantibody. Finally, in cases of suspected AIH and negative markers by IFA-RTT and AgSTs, it is strongly recommended to perform ANCA study given that, in some cases, it could be the only positive biomarker.

Regarding the immunological diagnosis of PBC, it is recommended to perform IFA-HEp-2 in addition to IFA-RTT for ANA screening. Of special interest is the detection of AC-6 and AC-12 patterns (ICAP nomenclature). AgST are also useful to determine anti-gp210 and anti-Sp100 autoantibodies. It is recommended to screen the AMA-M2 using IFA-RTT and IFA-HEp-2 for the detection of the AC-21 pattern. As for the AMA pattern on RTT, it is important to report it as AMA-M2, if this is the one detected. The remaining AMA patterns can be reported as atypical or different from AMA-M2 and not specifically associated with PBC. Furthermore, it is important to perform AgST in parallel, which allows the detection of the E2 subunits of the PDC, OGDC, BCOADC and E1 and E3BP complexes of PDC. The last two specificities can be included using native PDC. In some cases, where autoantibodies anti-BCOADC are present, the separate use of the E2 subunits may slightly increase sensitivity. Finally, in positive cases of AMA-M2 by IFA-RTT, they will also stain the GPC, thus the result of anti-GPC autoantibodies by IFA-RTT is not evaluable. Therefore, in case of suspicion of autoimmune gastritis, anti-GPC autoantibodies study should be performed using AgST for gastric anti-ATPase autoantibodies.

Although ANCA can appear in cases of AIH and supports its diagnosis in the absence of other autoantibodies, these were only determined by some laboratories. Discrepant results were obtained for AIH and pre-PBC patients and consensus was only reached for the patient with PSC diagnosis. In this last case, most participants reported this sample as an aANCA. The assessment of this kind of samples, that are positive on ethanol-fixed and negative on formalin-fixed neutrophils, is difficult because the presence of non-organ specific autoantibodies can mask IFA results. Beta-tubulin isotype 5 has been reported as the main target for aANCA in autoimmune liver and inflammatory bowel diseases [14] but these results have not been replicated by other authors and commercial kits are not currently available. Therefore, more studies are needed to clarify the role of ANCA in the immunological diagnosis of AILD.

Regarding anti-Ro52 autoantibodies, given that most laboratories used the same AgST, the results obtained could be biased. However, the DB technique seems to have good sensitivity and specificity. The poor correlation between anti-Ro52 autoantibodies detection methods may be associated with lower titres or differences in major epitope recognition between samples of Systemic Lupus Erythematosus, Sjögren Syndrome and Systemic Sclerosis or PBC patients [62]. Although the association between anti-SLA and anti-Ro52 autoantibodies in AIH patients has been described [63], both SLA positive patients included in the Autoimmunity Workshop GEAI-SEI 2020 were Ro52 negative. Even though it is still controversial and not specific for AILD, both Ro52 and centromere specificities may be considered AILD-associated and could be useful to enhance the diagnosis and prognostic accuracy, especially in seronegative patients.

The results of this nationwide work could constitute the basis for updating the AILD diagnostic algorithm proposed by GEAI-SEI in 2018.

## 5. Conclusions

In the case of suspected autoimmune liver disease, it is highly recommended to perform:Indirect immunofluorescence assays on rat triple tissue (IFA-RTT) and also on HEp-2 cells (IFA-HEp-2). The latter at least in the case of primary biliary cholangitis.Antigen-specific techniques, at least to detect the presence of anti-SLA autoantibodies, since it does not present a detectable pattern by IFA-RTT nor IFA-HEp-2, but ideally to confirm or rule out the presence of all autoantibodies whenever there is clinical suspicion.Study of anti-neutrophil cytoplasmic autoantibodies by IFA, if there is suspicion of autoimmune hepatitis or primary sclerosing cholangitis and the study performed at points 1 and 2 is negative.

## Figures and Tables

**Figure 1 diagnostics-12-00697-f001:**
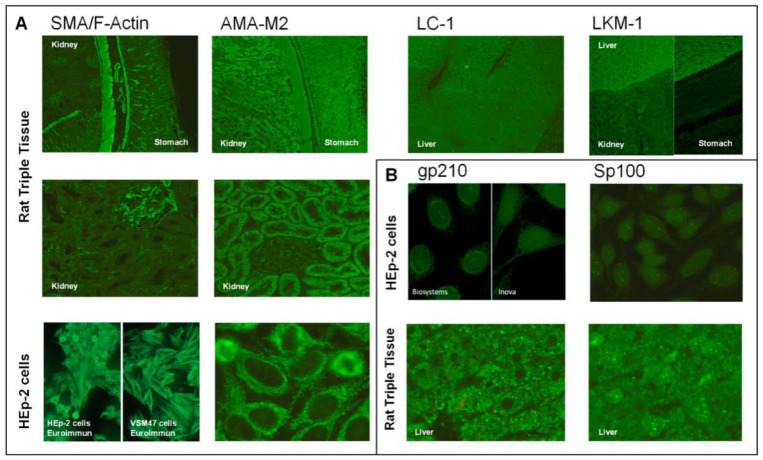
Indirect immunofluorescence patterns on rat triple tissue or HEp-2 cells of the main autoantibodies in AIH and PBC. Main anti-cytoplasmic (**A**) and anti-nuclear (**B**) patterns associated with AIH and PBC are shown. Abbreviations: AMA, anti-mitochondrial autoantibodies; gp210, glycoprotein 210; LC-1, liver cytosol-1; LKM-1, liver-kidney microsomal type-1; SMA, smooth muscle autoantibodies.

**Figure 2 diagnostics-12-00697-f002:**
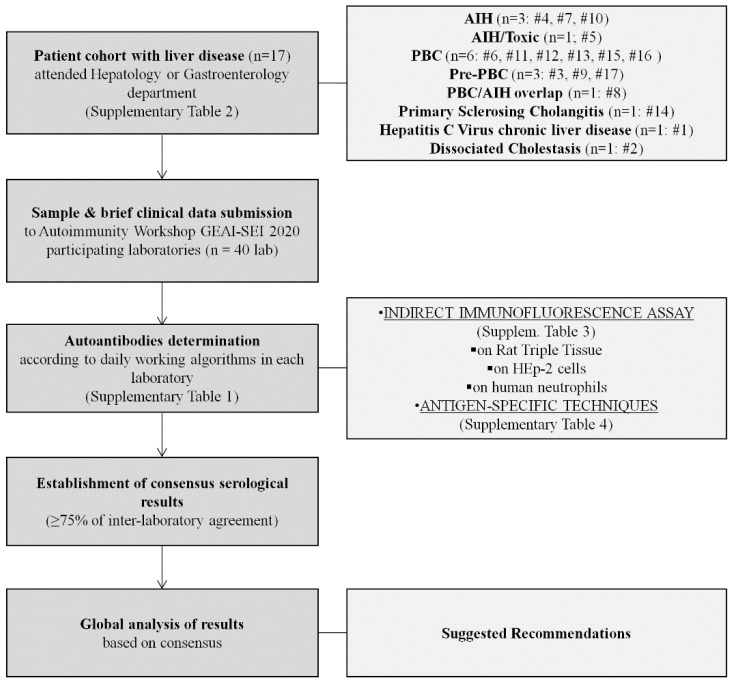
Flow-chart of Autoimmunity Workshop GEAI-SEI 2020. Abbreviations: AIH, autoimmune hepatitis; PBC, primary biliary cholangitis.

**Figure 3 diagnostics-12-00697-f003:**
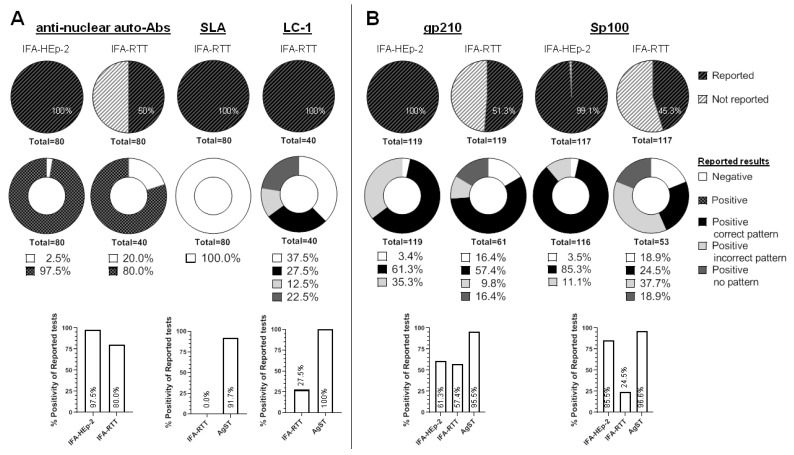
Summary of results obtained for associated anti-nuclear, anti-SLA and LC-1 autoantibodies. Results were obtained for each AIH- (**A**) and PBC- (**B**) associated autoantibody specificity employing IFA and/or AgST. Abbreviations: AgST, antigen-specific techniques; gp210, glycoprotein 210; IFA-HEp-2, indirect immunofluorescence assays on HEp-2 cells; IFA-RTT, indirect immunofluorescence assays on rat triple tissue; LC-1 liver cytosol-1; SLA, soluble liver antigen.

**Figure 4 diagnostics-12-00697-f004:**
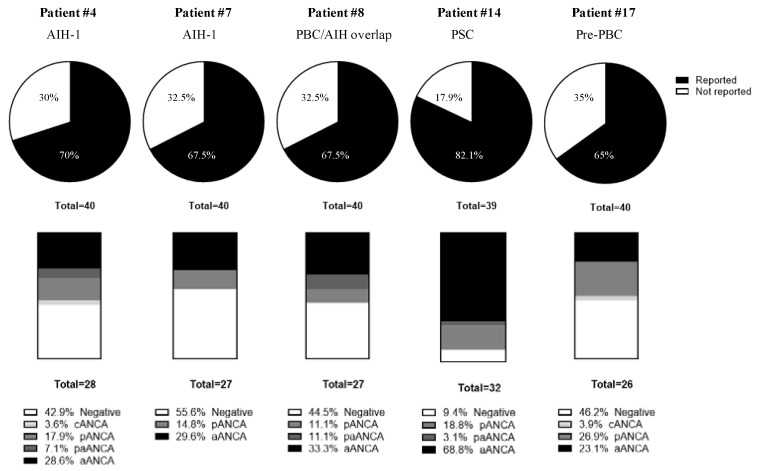
ANCA results in the five ANCA positive patients. Proportion of reported results for ANCA test and proportion of ANCA patterns for the reported results. Abbreviations: aANCA, atypical ANCA; ANCA, anti-neutrophil cytoplasm antibodies; cANCA, cytoplasmic ANCA; paANCA, perinuclear/atypical ANCA; pANCA, perinuclear ANCA; AIH-1, autoimmune hepatitis type 1; PBC, primary biliary cholangitis; PSC, primary sclerosing cholangitis.

**Table 2 diagnostics-12-00697-t002:** Description of anti-SMA/F-actin autoantibodies results in AIH patients.

			Indirect Immunofluorescence Assay			Antigen-Specific Techniques						
			Rat Triple Tissue		HEp-2 Cells							
**anti-SMA/F-Actin autoantibodies**	Pt	Diagnosis	SMA	SMA/F-Actin	AC-15/16/17	Total Performed	IFA EI VSM-47	DB AP	DB OR	DB DT	ELISA WI	Unknown
			% (n)	% (n)	% (n)	% (n)	% (n)	% (n)	% (n)	% (n)	% (n)	% (n)
	5	Toxic AIH	2	0	0	33	0%	0	0	0	0	0
			(1/39)	(0/39)	(0/38)	(13/39)	(0/2)	(0/2)	(0/1)	(0/4)	(0/2)	(0/2)
	4	AIH-1	5	0	0	33	0	0	0	0	33	0
			(2/40)	(0/40)	(0/38)	(13/39)	(0/2)	(0/2)	(0/0)	(0/4)	(1/3)	(0/2)
	7	AIH-1	93	13	13	64	50	100	0	40	100	20
			(37/40)	(5/40)	(5/39)	(25/40)	(4/8)	(2/2)	(0/1)	(2/5)	(4/4)	(1/5)
	2	Dissociated cholestasis	15	0	0	33	50	0	0	0	0	0
			(6/39)	(0/39)	(0/39)	(13/39)	(1/2)	(0/2)	(0/1)	(0/4)	(0/2)	(0/2)
	1	Chronic hepatopathy	75	7	18	51	66	100	100	66	100	20
			(29/39)	(3/39)	(7/39)	(20/39)	(4/6)	(2/2)	(1/1)	(2/3)	(3/3)	(1/5)
	10	AIH-2	0	0	2	33	0	0	0	0	0	0
			(0/40)	(0/40)	(1/38)	(13/39)	(0/2)	(0/2)	(0/1)	(0/4)	(0/2)	(0/2)
	8	PBC/AIH-1	93	25	36	66	100	50	100	100	100	50
			(37/40)	(10/40)	(14/39)	(26/40)	(9/9)	(1/2)	(1/1)	(4/4)	(4/4)	(3/6)

Percentage and number of patients with positive results for anti-SMA/F-actin autoantibodies using each test according to the method used. Percentages between 0–25% are shown in white; between 26–50% in light orange; between 51–75% in medium orange and between 76–100% in dark orange. Antigen-specific tests are also described according to the method used and the commercial house contracted for each laboratory. Non-reported commercial house is classified as unknown. Abbreviations: AC-15, cytoplasmic fibrillar linear; AC-16, cytoplasmic fibrillar filamentous; AC-17, cytoplasmic fibrillar segmental, according to ICAP nomenclature; DB, dot–blot; DT, D-Tek; EI, euroimmun; IFA-VSM47, indirect immunofluorescence assay on VSM47 line cell; OR, orgentec; Pt, patients.

**Table 3 diagnostics-12-00697-t003:** Description of AMA results in PBC, pre-PBC and PBC/AIH-1 overlap patients.

		Anti-Mitochondrial Autoantibodies										Anti-Gastric Parietal Cells Autoantibodies				
		Indirect Immunofluorescence Assay				Antigen-Specific Techniques						Rat Triple Tissue				Other Methods
		Rat Triple Tissue		HEp-2 Cells												
Pt	Diagnosis	AMA % (n)	AMA-M2 % (n)	AC-21 % (n)	Total % (n)	AMA Native % (n)	AMA-M2 % (n)	M2-3E2 % (n)	PDC-E2 % (n)	BCOADC E2 % (n)	OGDC E2 % (n)	Negative % (n)	Positive % (n)	Not Valuable % (n)	Total % (n)	% (n)
11	PBC	100	15	83	100	100	100	100	100	0	0	77	5	18	5	0
		(40/40)	(6/40)	(33/40)	(40/40)	(6/6)	(32/32)	(29/29)	(6/6)	(0/6)	(0/6)	(30/39)	(2/39)	(7/39)	(2/39)	(0/2)
12	PBC	100	15	85	100	100	97	100	100	100	0	69	15	15	8	100
		(40/40)	(6/40)	(34/40)	(40/40)	(7/7)	(30/31)	(28/28)	(6/6)	(6/6)	(0/6)	(27/39)	(6/39)	(6/39)	(3/39)	(3/3)
13	PBC	77	10	72	100	100	100	100	100	0	0	82	8	11	3	0
		(30/39)	(4/39)	(28/39)	(39/39)	(7/7)	(32/32)	(29/29)	(6/6)	(0/6)	(0/6)	(31/38)	(3/38)	(4/38)	(1/38)	(0/1)
15	PBC	97	13	85	100	100	100	100	100	0	0	76	5	18	5	0
		(38/39)	(5/39)	(33/39)	(39/39)	(7/7)	(32/32)	(28/28)	(6/6)	(0/6)	(0/6)	(29/38)	(2/38)	(7/38)	(2/38)	(0/2)
16	PBC	100	15	85	100	100	100	100	100	100	0	76	5	18	5	0
		(39/39)	(6/39)	(33/39)	(39/39)	(7/7)	(31/31)	(28/28)	(6/6)	(6/6)	(0/6)	(29/38)	(2/38)	(7/38)	(2/38)	(0/2)
6	PBC	5	0	3	100	0	0	71	0	0	0	100	0	0	0	-
		(2/40)	(0/40)	(1/40)	(40/40)	(0/6)	(0/29)	(20/28)	(0/5)	(0/5)	(0/5)	(39/39)	(0/39)	(0/39)	(0/39)	
8	PBC/AIH-1	25	0	45	100	0	0	77	0	100	0	87	8	5	3	100
		(10/40)	(0/40)	(18/40)	(40/40)	(0/7)	(0/30)	(23/30)	(0/6)	(6/6)	(0/6)	(34/39)	(3/39)	(2/39)	(1/39)	(1/1)
3	Pre-PBC	33	5	8	100	0	0	0	0	50	0	87	10	3	0	-
		(13/40)	(2/40)	(3/40)	(40/40)	(0/5)	(0/32)	(0/28)	(0/4)	(2/4)	(0/4)	(34/39)	(4/39)	(1/39)	(0/39)	
9	Pre-PBC	67	8	26	100	14	28	59	0	100	0	32	58	11	18	100
		(26/39)	(3/39)	(10/39)	(39/39)	(1/7)	(9/32)	(17/29)	(0/6)	(6/6)	(0/6)	(12/38)	(22/38)	(4/38)	(7/38)	(7/7)
17	Pre-PBC	13	3	15	100	60	20	37	0	25	50	100	0	0	0	-
		(5/40)	(1/40)	(6/40)	(40/40)	(3/5)	(6/30)	(11/30)	(0/4)	(1/4)	(2/4)	(39/39)	(0/39)	(0/39)	(0/39)	

Percentage and number of patients with positive results for AMA by each test according to the method used. Percentages between 0–25 are shown in white; between 26–50 in light orange; between 51–75 in medium orange and between 76–100 in dark orange. Data of gastric parietal cells autoantibodies are also presented. Antigen-specific tests are also described according to the antigen included in each test. Abbreviations: AC-21, cytoplasmic reticular/AMA pattern according to ICAP nomenclature; AMA, anti-mitochondrial autoantibodies; AMA-M2, bovine AMA that contain mainly E2-PDC-E2; AMA native, bovine/human native AMA antigen; AP, Alpha Dia, BCOADC-E2, E2-subunit of branched chain 2-oxo-acid dehydrogenase complex; M2-3E2, fusion peptide containing the three E2-subunits; OGDC-E2, E2-subunit of oxoglutarate dehydrogenase complex; PDC-E2, E2-subunit of pyruvate dehydrogenase complex; Pt, patients.

## Data Availability

Anonymised data not published within this article will be made available by request from qualified investigators.

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
