# Peer review of "Working Algorithms and Detection Methods of Autoantibodies in Autoimmune Liver Disease: A Nationwide Study"

_diagnostics, 2022, doi:10.3390/diagnostics12030697_

Round 1
Reviewer 1 Report
This is well written manuscript with interesting outcomes. Data curation, analysis, and execution were performed nicely and the findings were discussed adequtaly based on the data present in hand. Methodologies were described adequately. however, i suggest to add follwing minor things into the methodology section of the manuscript.
# The authors must add the details of kit and antibodies (catalogue no. company name, antibody host species type) they did use in the present study.
# The authors need to provide a brief description of the two immunoassays they did employ in the present manuscript.
Author Response
Thank you very much for your comments.
- Regarding the details about the commercial kits employed, we add the catalogue number of the most used ones in Suppl Table 4.
- Regarding the suggestion to include a brief description of the two main techniques (Immunofluorescence assay and Antigen specific techniques), given that they are basic well known techniques for laboratory specialists, we have considered more appropriate to reference the main laboratory book in immunological techniques (Manual of Molecular and Clinical Laboratory Immunology, 8th Edition). In these Chapters, 87, 88 and 94, the technical procedure is explained in detail.
Reviewer 2 Report
Muñoz-Sánchez et al. reported that Working Algorithms and Detection Methods of Auto-Antibodies in Autoimmune Liver Disease: A Nationwide Study. Manuscript is too immature to understand.
- In abstract section and text, do not use ALD, AST or auto-Abs. In general, ALD and AST mean alcoholic liver disease and aspartate aminotransferase, respectively.
- “anti-Liver Citosol-1”, Liver cytosol specific antibody type 1 (anti-LC1)??
- 464, 498 Conclusion??
- Authors should ask native English speaker to edit their manuscript before submission.
Author Response
First of all, we appreciate the time you have spent reviewing our manuscript.
Taking into account both your suggestions and those made by the other reviewers, we have tried to improve the present work.
Regarding the specific changes suggested:
- As you recommended, we have included more suitable abbreviations to refer to the following terms:
- Autoimmune liver disease: AILD instead of ALD
- Antigen specific technique: AgST instead of AST
Since, as you mention, ALD and AST are widely used terms referring to alcoholic liver disease and aspartate aminotransferase, respectively.
- In relation to auto-Abs: from an immunological point of view, since it is an autoimmune response, we believe it is more appropriate to keep the term auto-Antibodies (auto-Abs) as opposed to allo-antibodies present in other diseases. We use the plural because the autoimmune response is polyclonal.
- We have corrected the misspelling “anti-Liver Citosol-1” by “anti-Liver Cytosol-1” in the Introduction.
- We have slightly modified the paragraphs corresponding to the conclusions of the anti-gp210 and anti-Sp100 auto-Abs, according to your suggestion.
- Regarding the quality of the English language, we have made a review and introduced the corresponding changes.
Reviewer 3 Report
The present manuscript “Working Algorithms and Detection Methods of Auto-Antibodies in Autoimmune Liver Disease: A Nationwide Study” by Muñoz-Sánchez et al. is a very nice article overall. It reports of the nationwide Spanish study of the results of the 2020 Autoimmunity Workshop and to provides useful information to clinicians and laboratory specialists to improve the diagnosis of auto-Abs detection in autoimmune liver diseases. These were three differentiated entities including AIH, PBC and PSC, which are characterized by enhanced inflammation and progressive liver fibrosis. Since there were no consensus guidelines on working algorithms as well as on the most appropriate methods that should be used to detect these auto-Abs. This fact caused significant differences among autoimmunity laboratories. This study aimed for the first time to compare the working algorithms and the methods used by an important number of autoimmunity laboratories (n=40) nationwide in the context of autoimmune liver diseases.
This manuscript is written in good English, is comprehensive and has a good didactic structure.
Nonetheless, I suggest to add EASL/AASLD guidelines for AIH, PBC and PSC like these for AIH:
- European Association for the Study of the, L. EASL Clinical Practice Guidelines: Autoimmune hepatitis. Journal of hepatology 63, 971-1004, doi:10.1016/j.jhep.2015.06.030 (2015).
- Mack, C. L. et al. Diagnosis and Management of Autoimmune Hepatitis in Adults and Children: 2019 Practice Guidance and Guidelines From the American Association for the Study of Liver Diseases. Hepatology 72, 671-722, doi:10.1002/hep.31065 (2020).
Author Response
Thank you very much for highlighting the strengths of our work.
Regarding your suggestions, we fully agree and have introduced the EASL/AASLD guidelines for AIH, PBC and PSC in the manuscript (Ref 4, 13, 38, 39).
Round 2
Reviewer 2 Report
- Do not use “auto-Ab”. Please use autoantibody.
- Please ask native English speaker to edit your manuscript before submission.
Author Response
Dear reviewer,
We have introduced last modifications following your suggestions:
1) We have changed the abbreviation "auto-Ab” for autoantibody/ies.
2) The manuscript has been revised by a native English speaker.
Thank you in advance.